# ENHANCING LLM ROBUSTNESS TO PERTURBED INSTRUCTIONS: AN EMPIRICAL STUDY

**Aryan Agrawal**[∗]**, Lisa Alazraki**[∗]**, Shahin Honarvar, Marek Rei**
Imperial College London

## ABSTRACT

Large Language Models (LLMs) are highly vulnerable to input perturbations, as even a small prompt change may result in a substantially different output. Existing methods to enhance LLM robustness are primarily focused on perturbed data samples, whereas improving resiliency to perturbations of task-level instructions has remained relatively underexplored. In this work, we focus on character- and word-level edits of task-specific instructions, which substantially degrade downstream performance. We experiment with a variety of techniques to enhance the robustness of LLMs, including self-denoising and representation alignment, testing different models (Llama 3 and Flan-T5), datasets (CoLA, QNLI, SST-2) and instructions (both task-oriented and role-oriented). We find that, on average, self-denoising—whether performed by a frozen LLM or a fine-tuned model—achieves substantially higher performance gains than alternative strategies, including more complex baselines such as ensembling and supervised methods. We share our data and code at https://github.com/ary4n99/llm-robustness.

## 1 INTRODUCTION

Despite achieving impressive performance in increasingly sophisticated tasks (Nori et al., 2023; Roemmele & Gordon, 2024), LLMs remain sensitive to input perturbations (Wang et al., 2024b). While human performance in natural language tasks is resilient to small alterations in the problem description (Walkington et al., 2019), LLMs have consistently been observed to shift their output dramatically even with minor changes to the input (Moradi & Samwald, 2021; Wang et al., 2022c; Alazraki et al., 2023; Gulati et al., 2024; Honarvar et al., 2025). Prior literature extensively investigates improving LLM robustness when individual data samples are perturbed (Hu et al., 2024; Wang et al., 2024a; Chen et al., 2025). On the other hand, methods for handling perturbations of task-level instructions (i.e., fixed templates that are combined with each data point to help solve a task) are less researched.

Intuitively, instruction quality and readability have a considerable impact on performance: perturbing the instruction can potentially lead the LLM to misunderstand the task and fail on all samples. Indeed, prior work finds that LLM proficiency varies widely when instructions are paraphrased (Mizrahi et al., 2024; Zhu et al., 2024) or individual words are replaced, added or removed (Gu et al., 2023; Zhu et al., 2024). Similarly, Sun et al. (2024) study the performance of instruction-tuned LLMs when test-time instructions are phrased differently from the training data, and observe substantial degradation across different models and tasks. As a solution, they propose aligning the internal model representations of the rephrased instructions to those of the original ones.

In this work, we investigate a range of methods—both prompt-based and fine-tuned—for enhancing LLM robustness to perturbed instructions in classification tasks. We focus on word- and character-level perturbations, as these have been found to cause the greatest performance decline (Zhu et al., 2024). We assess each method on a combination of six instructions, two types of perturbations, three datasets, and two base models. Our experiments show that LLMs are particularly effective at self-denoising instructions, especially when the process is done iteratively.

Our main findings are as follows: (1) iterative self-denoising—whether carried out by a fine-tuned model or the base LLM—prevents a considerable portion of the performance drop caused by using

---

∗Equal contribution. Correspondence to contact@aryanagrawal.com, lisa.alazraki20@imperial.ac.uk.

(a) Non-perturbed.      (b) DeepWordBug.      (c) TextFooler.

Figure 1: Example perturbations of an instruction for sentiment classification, shown in (a). The perturbation can be at the character level, as shown in (b), or at the word level, as shown in (c).

perturbed instructions in classification tasks; (2) self-denoising in general is far more effective than other methods including instruction ensembling and hidden representation alignment; (3) other denoising strategies, such as perplexity smoothing, are not as successful. In fact, they tend to decrease performance further.

## 2 METHODS

We compare several methods to enhance LLM robustness to instruction perturbations. Namely, (iterative) self-denoising, perplexity smoothing, instruction ensembling, representation alignment. We have chosen these methods because they represent intrinsically different techniques to improve robustness. Further implementation details and hyperparameters can be found in Appendix A.

### 2.1 SELF-DENOISING

In self-denoising, we ask the LLM to unperturb a given instruction, given a meta-prompt and a set of in-context learning (ICL) examples (shown in Appendix B). Throughout the paper, we abbreviate this method as SD. We additionally explore variants of this method, described below.

**Iterative self-denoising (SDi)** This variant progressively unperturbs an instruction over multiple calls to the LLM, using the same meta-prompt and examples. If the instruction has not changed from the previous iteration, the process is stopped. Otherwise, the process will stop after five iterations.

**Supervised fine-tuned self-denoising (SFT-SD)** We fine-tune an LLM for the task of unperturbing instructions. We perform parameter-efficient tuning by adding LoRA modules (Hu et al., 2022) to the value and query projection layers of the frozen LLM. We create a novel training dataset, AdvMix, containing 2,900 pairs of perturbed and unperturbed sequences extracted from AdvGLUE (Wang et al., 2022b) and PromptBench (Zhu et al., 2024). Details of AdvMix are given in Appendix C. During both training and inference, the model observes the self-denoising meta-prompt and ICL examples. Note that the fine-tuned model can be applied in an iterative fashion at test time (SFT-SDi).

### 2.2 PERPLEXITY SMOOTHING

Inspired by randomised smoothing methods (Cohen et al., 2019; Gietz & Kalita, 2024; Robey et al., 2024; Zhang et al., 2024), we build a framework that minimises perplexity (PPL) as a proxy metric for the integrity of an instruction. Firstly, we rank words within an instruction by importance (leaving out stop words and class labels), where the importance of a word $w_j$ is a function of the change in PPL of the instruction when $w_j$ is deleted. We then mask the $n$ top-ranked $w_j$ with a `[MASK]` token, and adopt a masked language model to generate $k$ candidate words to fill the mask. Having generated $k$ variants of the instruction containing each candidate word, we select the $\beta$ lowest-PPL variants and repeat the procedure masking the next-ranked word. We take the final, PPL-smoothed instruction resulting from this beam search process as the denoised instruction.

### 2.3 INSTRUCTION ENSEMBLING

We ensemble $n$ variations of an instruction, each obtained by sampling with temperature from an LLM, using the same meta-prompt and ICL examples as in the self-denoising pipeline. We run

---

**Algorithm 1:** Greedy Search for Optimal Perturbation

---

**Input:** input instruction $i$, continuous goal function $\mathcal{G}$, set of transformations $\mathcal{T}$, set of constraints $\mathcal{C}$, query limit $q_{max}$
**Output:** optimal perturbed instruction $i^*$

$i^* \leftarrow i$
$q \leftarrow 0$
**while** $q < q_{max}$ **do**
    $\mathcal{I} \leftarrow \{\,\}$
    **for** $T \in \mathcal{T}$ **do**
        $i' \leftarrow T(i^*)$
        **if** $C(i')$ *is satisfied* $\forall C \in \mathcal{C}$ **then**
            $\mathcal{I} \leftarrow \mathcal{I} \cup \{i'\}$

    **if** $\mathcal{I} = \emptyset$ **then**
        `// No valid transformations from current` $i^*$
        **break**
    $n \leftarrow \min(q_{max} - q, |\mathcal{I}|)$
    $i^* \leftarrow \underset{i' \in \mathcal{I}_{1:n}}{\arg\max}\, \mathcal{G}(i')$
    $q \leftarrow q + |\mathcal{I}_{1:n}|$
**return** $i^*$

---

inference on each data sample using all $n$ variations, and select the final classification label by majority vote.

## 2.4 REPRESENTATION ALIGNMENT

For comparison, we implement a framework to align the hidden representation of the perturbed instruction to that of the non-perturbed one, similar to Sun et al. (2024). Given a dataset $\mathcal{D} = \{(i_j, i'_j)_{1 \leq j \leq N}\}$ containing pairs of unperturbed and perturbed instructions, we add LoRA adapter modules to a frozen LLM and train with the objective to minimise the cosine distance between $h(i_j)$ and $h(i'_j)$, where $h(i)$ is the hidden representation of $i$ at the middle layer of the LLM. We use AdvMix for training. We choose the middle layer as a trade-off between capturing basic semantics (potentially useful for simpler perturbations, such as character-level edits) and representing contextual meaning (which may be relevant for more complex, word-level perturbations).

## 3 EXPERIMENTS

We run all experiments with two well-known open-weight LLMs—Llama 3 8B Instruct (Dubey et al., 2024) and Flan-T5 Large (Chung et al., 2024). We refer to these models as Llama 3 and Flan-T5.

### 3.1 PERTURBATIONS

Given an instruction $i$, we obtain its perturbed version using a framework adapted from Morris et al. (2020)'s TextAttack. We greedily search for an optimal perturbation among the space of all possible perturbations $\mathcal{I}$, given a goal function $\mathcal{G}$. The search strategy is illustrated in Algorithm 1. Note that unlike in Morris et al. (2020), we do not implement early stopping upon $\mathcal{G}(i')$ reaching a threshold. The optimal perturbation can thus be defined as

$$i^* = \arg\max_{i' \in \mathcal{I}} C(i')\mathcal{G}(i'),$$

where $C(\cdot)$ is an indicator function, returning 1 when adhering to the constraints. In our case, the goal of the attack is to maximise the performance drop produced by the perturbed instruction. The

Table 1: Performance Drop Rate (PDR) obtained with perturbed instructions, aggregated by perturbation type, model and dataset. Lower PDR scores are better. For each method, we also report the average PDR improvement, i.e., the overall percentage change in PDR from the base LLM.

| | PDR (↓) | | | | | | | Avg. PDR improvement (↑) |
| | Perturbation | | Model | | Dataset | | | |
| | TextFooler | DeepWord-Bug | Llama 3 | Flan-T5 | CoLA | QNLI | SST-2 | |
|---|---|---|---|---|---|---|---|---|
| Base LLM | 0.174 | 0.077 | 0.192 | 0.059 | 0.102 | 0.140 | 0.134 | – |
| PPL smoothing | 0.182 | 0.110 | 0.214 | 0.078 | 0.115 | 0.150 | 0.172 | −16.3% |
| Instr. ensembling | 0.130 | 0.037 | 0.142 | 0.026 | 0.071 | 0.094 | 0.086 | 33.3% |
| Repr. alignment | 0.113 | 0.053 | 0.125 | 0.041 | **0.052** | 0.117 | 0.080 | 33.8% |
| SD | 0.130 | 0.016 | 0.122 | 0.025 | 0.057 | 0.085 | 0.077 | 41.7% |
| SDi | 0.125 | **0.015** | 0.119 | 0.021 | 0.053 | 0.091 | 0.065 | 44.3% |
| SFT-SDi | **0.072** | 0.030 | **0.082** | **0.021** | 0.055 | **0.062** | **0.036** | **59.2%** |

constraints are that stop words and class labels must remain unperturbed. We choose perturbations that have been found to cause substantial performance degradation in previous literature (Zhu et al., 2024). These include character-level substitutions, insertions and deletions (Figure 1b) obtained with DeepWordBug (Gao et al., 2018), and word replacements by counter-fitted GloVe embeddings (Pennington et al., 2014) (Figure 1c) obtained using TextFooler (Jin et al., 2020).

## 3.2 DATASETS

We evaluate the LLMs on three classification tasks from the GLUE benchmark (Wang et al., 2018). On these, base models achieve strong results, yet perturbing the instruction causes substantial performance loss (Zhu et al., 2024). The tasks are: (1) CoLA (Warstadt et al., 2019), which consists of 1k texts labelled as 'acceptable' or 'unacceptable' from a grammatical standpoint, (2) QNLI (Rajpurkar et al., 2016), a natural language inference dataset containing 5.5k samples, (3) SST-2 (Socher et al., 2013), comprising 1.8k text samples for binary sentiment analysis extracted from movie reviews. For each test set, we use six zero-shot instructions from the PromptBench library (Zhu et al., 2024), split among task-oriented and role-oriented. Instructions are shown in Appendix D.

## 3.3 METRIC

We evaluate the efficacy of the methods with Performance Drop Rate (PDR) (Zhu et al., 2024). This metric measures the degradation in performance (i.e., classification accuracy) of an LLM under a perturbation, hence lower PDR values are better. Given a perturbation $P$, an instruction $i$, a robustness augmentation $\Phi$, a base model $f_\theta$, and a dataset of samples $\mathcal{D}_s = \{(x_j, y_j)_{1 \leq j \leq N}\}$, we compute the PDR as

$$\mathrm{PDR}(P, i, \Phi, f_\theta, \mathcal{D}_s) = 1 - \frac{\sum_{j=1}^{N} \mathbb{1}\{\Phi(f_\theta, P(i), x_j) = y_j\}}{\sum_{j=1}^{N} \mathbb{1}\{\Phi(f_\theta, i, x_j) = y_j\}}.$$

Note that the performance discrepancy between the base LLMs and their $\Phi$ augmented versions is negligible when the instruction is clean (see Table 3 in Appendix E). This holds true for all the methods in Section 2, as none of them make substantial changes to a non-perturbed instruction.

## 3.4 RESULTS AND ANALYSIS

Table 1 displays the PDR scores for each method, aggregated by perturbation type, model and dataset. We find that SFT-SDi is the best performing strategy overall, with 59.2% average PDR improvement. Generally, self-denoising achieves low PDR scores, even in the non-fine-tuned iterative

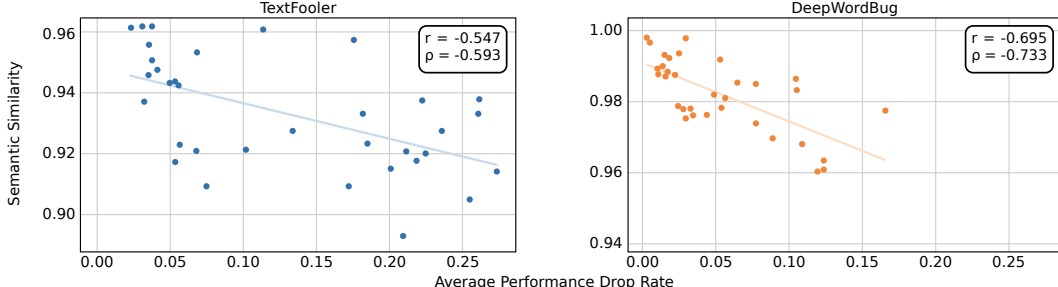

Figure 2: PDR and semantic similarity for TextFooler and DeepWordBug, averaged across models, datasets and instruction variants. For semantic similarity, we use the cosine similarity between the 4096-dimensional sentence embeddings encoded by E5 Mistral (Wang et al., 2024c). We choose this model since, at the time of writing, it achieves leading performance on the Massive Text Embedding Benchmark (MTEB) (Muennighoff et al., 2023), which is designed to evaluate the quality of text embeddings on a variety of tasks, including semantic similarity and text classification.

setting (SDi, obtaining 44.3% avg. PDR improvement) and the vanilla SD setting (41.7%). We observe that instruction ensembling and representation alignment perform fairly similarly overall (33.3% and 33.8% avg. PDR improvement, respectively). We also find that PPL smoothing results in a performance decrease. This method increases the PDR over the base LLM in all cases, which reflects a negative PDR improvement.

It is worth noting that the lower PDR improvement given by representation alignment and instruction ensembling is mostly due to their PDR scores on DeepWordBug perturbations (.053 and .037 respectively, vs .016 for SD). On TextFooler, on the other hand, representation alignment achieves better PDR than both SD and SDi (though not SFT-SDi), while ensembling obtains a comparable PDR. In Figure 2, we analyse the effects of both perturbation types. We observe that TextFooler produces instructions that are semantically less similar to the original compared to DeepWordBug. This suggests that representation alignment and ensembling are effective when perturbed instructions substantially diverge semantically from the original, but they may be unsuitable for more subtle perturbations. Finally, we observe that semantic similarity is negatively correlated with PDR, with a stronger negative correlation for DeepWordBug perturbations ($r = -0.695$, $\rho = -0.733$) compared to TextFooler ($r = -0.547$, $\rho = -0.593$), suggesting that greater semantic deviation between the perturbed and original instructions leads to higher performance degradation.

## 4 CONCLUSION

We have investigated an extensive range of methods to enhance LLM robustness to perturbed instructions, across multiple models, datasets, perturbations and instruction templates. We have found that self-denoising—even in its simplest form—performs better on average than other methods. This highlights the ability of LLMs to self-correct perturbations to their instructions. We also observed that perplexity smoothing is completely ineffective at reducing PDR, causing instead a further loss in performance. Our empirical study lays substantial groundwork for the underexplored domain of LLM robustness to instruction perturbations, highlighting the most promising methods. Future research can further build upon these strategies, potentially investigating tasks beyond classification, larger model sizes and more complex perturbations such as semantic paraphrasing.

## ACKNOWLEDGMENTS

We would like to thank Thomas Mensink for the valuable advice he offered throughout this work, from its inception to the write-up. We also thank Fantine Huot, who provided many insightful comments on the first draft of this paper.

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

# A IMPLEMENTATION DETAILS

## A.1 SELF-DENOISING

For vanilla self-denoising (SD) and iterative self-denoising (SDi), we use greedy decoding with temperature $t = 0$.

In Table 2 we detail the hyperparameters used for training the LoRA modules in the fine-tuned self-denoising pipeline (SFT-SD) on AdvMix. The same hyperparameters are used for both Llama 3 and Flan-T5. Note that we use the hyperparameter combination from Dettmers et al. (2023), since this has been shown to generalise well to a wide range of tasks.

## A.2 PERPLEXITY SMOOTHING

All PPL scores in perplexity smoothing are computed using GPT-2 (Radford et al., 2019). Candidate substitute words for each [MASK] token are found using DistilRoberta Base[1]. In our experiments, we set $n = 10$ (i.e., we mask the top ten most important words). We also set the same beam width $\beta$ as the number of candidates $k$, i.e. $k = \beta = 5$, thus performing best-first search.

## A.3 INSTRUCTION ENSEMBLING

For instruction ensembling, we sample $n$ options from the LLM with temperature $t = 1$, and set $n = 5$. As the classification tasks in our experimental setup are binary, this value of $n$ ensures that it is always possible to take the majority label as the final classification label.

## A.4 REPRESENTATION ALIGNMENT

We use a siamese model implementation (Chen et al., 2020; Xu et al., 2023; Sun et al., 2024) to align the hidden representations of the perturbed instructions to those of the non-perturbed instructions. We align representations at the layer $l$, where $l$ is chosen to be the middle layer of the LLM. For Llama 3 8B Instruct (32 decoder layers), we set $l = 16$. For Flan-T5, we take advantage of the encoder-decoder architecture and set $l$ to be last hidden layer of the encoder block. The siamese network is trained on the instruction pairs in AdvMix using LoRA modules (Hu et al., 2022) at the value and query projection layers of the LLM. The LoRA modules are disabled or enabled during each forward pass depending on whether the input consists of unperturbed or perturbed instructions, respectively. Since unperturbed and perturbed instructions may differ in token count, mean pooling is applied to their middle-layer hidden representations before computing the cosine distance loss.

The training hyperparameters—for both the Llama 3 and the Flan-T5 implementation—are shown in Table 2 (note that the same hyperparameters are used for training the SFT self-denoising models).

---

[1]https://huggingface.co/distilbert/distilroberta-base

Table 2: Hyperparameters for training the SFT self-denoising models and the representation alignment network. The hyperparameter combination is the same for both base LLMs (Llama 3 and Flan-T5).

| Hyperparameter | Value |
|---|---|
| LoRA $\alpha$ | 16 |
| LoRA $r$ | 64 |
| LoRA dropout | 0.1 |
| LoRA modules | $Q_{\text{proj}}, V_{\text{proj}}$ |
| Learning rate | 5e−5 |
| Batch size (effective) | 4 |
| Epochs | 10 |

## B   SELF-DENOISING META-PROMPT AND EXAMPLES

In Prompt 1, we show the meta-prompt and few-shot examples used at both training and inference in the self-denoising pipeline. All examples are extracted from MNLI (Williams et al., 2018), as these achieved the highest validation results across the different datasets, surpassing setups where the exemplar instructions were extracted from multiple diverse tasks.

---

**Prompt 1:** Meta-prompt and examples for self-denoising

```
Given a sentence which could be perturbed through an adversarial
attack, respond with the unperturbed sentence. Do not modify the
following words: {excluded_words}. Do not answer with anything
other than the unperturbed sentence.

Uncovering whether the made coupling of condemns revealed
entailment, neutral, or contradiction. Cope with 'entailment',
'neutral', or 'contradiction':

Identify whether the given pair of sentences demonstrates
entailment, neutral, or contradiction. Answer with 'entailment',
'neutral', or 'contradiction':

Specifies if the made coupling of condemns exposure entailment,
neutral, or contradiction. Reacting with 'entailment', 'neutral',
or 'contradiction':

Determine if the given pair of sentences displays entailment,
neutral, or contradiction. Respond with 'entailment', 'neutral',
or 'contradiction':

Can the ratio between the offered penalty be entailment, neutral,
or contradiction? Reactions with 'entailment', 'neutral', or
'contradiction':

Does the relationship between the given sentences represent
entailment, neutral, or contradiction? Respond with 'entailment',
'neutral', or 'contradiction':
```

## C  TRAINING DATA

We train the SFT-SD model and the representation alignment pipeline on AdvMix, a custom dataset containing 2,882 pairs of unperturbed and perturbed text sequences.

To create AdvMix, we extract 2,530 pairs of sequences (88% of the total) from the AdvGLUE benchmark (Wang et al., 2022b), which contains the original and perturbed versions of the samples in GLUE (Wang et al., 2018). Our AdvGLUE selection is split among Quora Question Pairs (QQP) (Sharma et al., 2019), Multi-Genre Natural Language Inference (MNLI) (Williams et al., 2018), and Recognizing Textual Entailment (RTE) (Dagan et al., 2005). Note that none of these datasets are used at test time.

Since AdvGLUE contains general text sequences rather than instructions, we augment AdvMix with 352 samples from PromptBench (Zhu et al., 2024), which contains perturbed and unpreturbed instruction prompts. Again, we are careful to use use datasets that do not appear among our testing data. We extract from Microsoft Research Paraphrase Corpus (MRPC) (Dolan & Brockett, 2005).

Note that both AdvGLUE and PromptBench include more than one perturbation for each clean text sequence. To aid the generalization of our train model, we include these different perturbations in AdvMix. We thus source sequences perturbed via four word-level attacks (BERT-Attack (Li et al., 2020), SememePSO (Zang et al., 2020), TextBugger (Li et al., 2019), TextFooler (Jin et al., 2020)) and seven sentence-level attacks (CheckList (Ribeiro et al., 2020), AdvFever (Thorne et al., 2019), SCPN (Iyyer et al., 2018), SemAttack (Wang et al., 2022a), T3 (Wang et al., 2020), StressTest (Naik et al., 2018), and DeepWordBug (Gao et al., 2018)).

## D  INSTRUCTIONS

We use six instruction variants for each dataset, equally split among *task-oriented* and *role-oriented* instructions.

### D.1  CoLA INSTRUCTIONS

---

**Task-Oriented**

```
Assess the following sentence and determine if it is grammatically
correct.  Respond with 'acceptable' or 'unacceptable':

Examine the given sentence and decide if it is grammatically sound.
Answer with either 'acceptable' or 'unacceptable':

Analyze the provided sentence and classify its grammatical correctness as
'acceptable' or 'unacceptable':
```

**Role-Oriented**

```
In your role as a grammar check tool, assess the following sentence
and classify it as 'acceptable' if it is grammatically correct or
'unacceptable' if it is incorrect:

As a grammar identification system, examine the provided sentence
and respond with 'acceptable' for grammatically correct sentences or
'unacceptable' for incorrect ones:

Functioning as a grammar evaluation tool, analyze the given sentence and
decide if it is grammatically correct, responding with 'acceptable' or
'unacceptable':
```

---

## D.2 QNLI INSTRUCTIONS

---

**Task-Oriented**

Given the question and context provided, determine if the answer can be inferred by choosing 'entailment' or 'not entailment'.

Based on the provided context and question, decide if the information supports the answer by responding with 'entailment' or 'not entailment'.

Please assess if the answer to the question can be derived from the given context by selecting 'entailment' or 'not entailment'.

**Role-Oriented**

As a language expert, assess if the given context entails the answer to the question and respond with 'entailment' or 'not entailment'.

In your role as a semantic evaluator, determine if the provided context justifies the answer to the question and answer with 'entailment' or 'not entailment'.

As a textual analyst, examine if the given context logically implies the answer to the question and indicate your decision with 'entailment' or 'not entailment'.

---

## D.3 SST-2 INSTRUCTIONS

---

**Task-Oriented**

Analyze the tone of this statement and respond with either 'positive' or 'negative':

Evaluate the sentiment of the given text and classify it as 'positive' or 'negative':

Please identify the emotional tone of this passage: 'positive' or 'negative'?

**Role-Oriented**

As a sentiment classifier, determine whether the following text is 'positive' or 'negative'. Please classify:

In the role of a sentiment analysis tool, respond with 'positive' or 'negative' to classify this statement:

Acting as a sentiment evaluator, identify if the given sentence is 'positive' or 'negative'. Classify:

---

# E   PERFORMANCE WITH NON-PERTURBED INSTRUCTIONS

In Table 3 we show the classification accuracy on non-perturbed instructions for each method. Scores are averaged across datasets (CoLA, QNLI, SST-2), underlying LLMs (Llama 3, Flan-T5) and instruction variants (six variants for each dataset). Note that the accuracy scores obtained by the augmented pipelines are within only 1% of that achieved using the base model implementation.

Table 3: Accuracy scores for each method, averaged across datasets and LLMs.

| Method | Avg. performance |
|---|---|
| Base LLM | 80.1 |
| PPL smoothing | 79.3 |
| Instruction ensembling | 80.0 |
| Representation alignment | 79.8 |
| SD | 80.0 |
| SDi | 80.0 |
| SFT-SDi | 79.1 |

