# OpenReview forum: "Enhancing LLM Robustness to Perturbed Instructions: An Empirical Study"
_ICLR.cc/2025/Workshop/BuildingTrust — BuildingTrust_

### Official Review · Reviewer_ag1J · 2025-03-01
**Comprehensive Approach to Enhancing LLM Robustness Against Perturbed Instructions**

**Rating:** 6
**Confidence:** 4

**Review:**

This paper investigates methods to enhance the robustness of large language models (LLMs) against perturbed instructions, focusing on character- and word-level edits that degrade downstream performance. The authors experiment with various techniques, including self-denoising, representation alignment, instruction ensembling, and perplexity smoothing, across two LLMs (Llama 3 and Flan-T5), three datasets (CoLA, QNLI, SST-2), and six instruction variants. Their findings indicate that iterative self-denoising (SFT-SDi) outperforms other methods, achieving an average improvement of 59.2% in Performance Drop Rate (PDR).

Strengths:
- The study addresses a critical issue in the deployment of LLMs: their sensitivity to perturbations in task instructions. This is particularly relevant for real-world applications where input variations are common.
- The authors conduct extensive experiments using multiple models, datasets, and perturbation types, ensuring a thorough evaluation of the proposed methods.
- The results clearly demonstrate the effectiveness of iterative self-denoising (SFT-SDi) over alternative approaches, providing actionable insights for improving LLM robustness.
- The paper highlights the potential of LLMs to self-correct perturbed instructions, which could lead to practical solutions for enhancing model reliability in noisy or adversarial environments.

Weaknesses:
- While the paper focuses on character- and word-level perturbations, it does not explore more complex or context-aware perturbations (e.g., semantic paraphrasing or adversarial attacks targeting meaning). Expanding the scope would strengthen the conclusions.
- The experiments are conducted on mid-sized LLMs (Llama 3 8B and Flan-T5 Large). It would be valuable to test these methods on larger, state-of-the-art models to assess scalability and generalizability.
- The paper does not compare the proposed methods with broader robustness-enhancing techniques, such as adversarial training or data augmentation. Including such comparisons would provide a more comprehensive perspective.
- The computational cost of the proposed methods, especially iterative self-denoising, is not discussed. This is important for understanding their feasibility in real-time applications.

The paper makes a valuable contribution by systematically evaluating methods to improve LLM robustness against perturbed instructions. The proposed iterative self-denoising approach demonstrates strong performance improvements, offering a promising direction for future research. However, the study could benefit from addressing the aforementioned limitations, particularly regarding perturbation complexity, computational efficiency, and scalability to larger models. With these refinements, the paper has the potential to significantly advance the field of LLM robustness.

---

### Official Review · Reviewer_UhdV · 2025-03-02
**A well-executed position study on how LLMs handle perturbations in task-specific instructions.**

**Rating:** 7
**Confidence:** 3

**Review:**

The paper addresses an important and underexplored problem: how LLMs handle perturbations in task-specific instructions. The empirical study provides insights into various strategies to enhance robustness.

The study is well-designed, testing different perturbation types (character and word level) on multiple datasets (CoLA, QNLI, SST-2) using two prominent models (LLaMA 3 and Flan-T5). In addition, it compares various robustness-enhancing techniques, including self-denoising (SD), perplexity smoothing, instruction ensembling, and representation alignment.

While the empirical findings are strong, the paper lacks a deeper theoretical justification for why self-denoising works significantly better than other methods. Besides, the paper focuses on classification tasks, but it would be valuable to test whether the findings hold for other NLP tasks such as text generation, translation, or question-answering. Ablation Studies on Self-Denoising are essential, e.g., How does self-denoising compare when using different model sizes or different fine-tuning strategies? Does self-denoising performance degrade with increasing instruction complexity?

This is a well-executed empirical position study with meaningful contributions to LLM robustness. The experimental setup is strong, and the findings are well-supported by the data. Future work could improve the theoretical justification.

---

### Decision · Program_Chairs · 2025-03-02

Accept